# Towards Interpretable Deep Reinforcement Learning with Human-Friendly Prototypes

**Eoin M. Kenny, Mycal Tucker & Julie A. Shah**
Computer Science & Artificial Intelligence Laboratory
Massachusetts Institute of Technology
{ekenny,mycal,julie_a_shah}@mit.edu

## Abstract

Despite recent success of deep learning models in research settings, their application in sensitive domains remains limited because of their opaque decision-making processes. Taking to this challenge, people have proposed various eXplainable AI (XAI) techniques designed to calibrate trust and understandability of black-box models, with the vast majority of work focused on supervised learning. Here, we focus on making an "interpretable-by-design" *deep* reinforcement learning agent which is forced to use human-friendly prototypes in its decisions, thus making its reasoning process clear. Our proposed method, dubbed Prototype-Wrapper Network (PW-Net), wraps around any neural agent backbone, and results indicate that it does not worsen performance relative to black-box models. Most importantly, we found in a user study that PW-Nets supported better trust calibration and task performance relative to standard interpretability approaches and black-boxes.

## 1 Introduction

Deep reinforcement learning (RL) models have achieved state-of-the-art results in Go (Silver et al., 2016), Chess (Silver et al., 2017), Atari (Mnih et al., 2015), self-driving cars (Kiran et al., 2021), and robotic control (Kober et al., 2013). However, the usage of these agents in truly sensitive domains is limited due to the opaque nature of such systems. Extracting a deep model's rationale in a human interpretable format remains a challenging problem, but doing so would be highly useful to troubleshoot an agent's actions and by extension its possible failure states (Hayes & Shah, 2017). One popular approach to do so is with *post-hoc* explanation methods, which give after-the-fact rationales for model predictions mostly through some form of saliency map (Bach et al., 2015) or exemplar (Kenny & Keane, 2021). However, whilst popular, these approaches may be incomplete or unsuitable for explanation (Slack et al., 2020; Zhou et al., 2022), and recent work has instead started to focus on *pre-hoc* interpretability (Rudin, 2019). The core idea behind this latter paradigm is to design inherently explainable models, so that you can clearly see and understand their decision-making process in such a way that you can calibrate user trust and predict the system's capabilities.

In this paper, we present (to the best of our knowledge) the first general, inherently interpretable, well performing, deep reinforcement learning (RL) algorithm that uses an intuitive exemplar-based approach for decision making. Specifically, we train a "wrapper" model called Prototype-Wrapper Network (PW-Net) that can be added to any pre-trained agent, which allows them to be interpretable-by-design, offering the same intuitive reasoning process as popular *pre-hoc* methods (Li et al., 2018). Crucially however, when using PW-Nets, the main advantages of *post-hoc* methods remain in-tact, in that the black-box model's performance is *not* lost, and it doesn't need to be retrained from scratch, which we show across multiple domains notoriously difficult for XAI.

## 2 Related Work

This paper builds upon recent work building prototype-based neural networks for interpretable supervised learning. Such networks are interpretable by design because they utilize these prototypes in their forward pass by classifying test instances based upon their proximity to these prototypes, thus allowing users to intuitively understand predictions. Perhaps the first notable example of this was by

Li et al. (2018) who learned prototypes in latent space which classified test instances using their $L_2$ distance to each prototype. Work in this area was followed up by Chen et al. (2019) who used image "parts" rather than the whole instance. This helped spawn many followup works in NLP (Ming et al., 2019), fairness (Tucker & Shah, 2022), and computer vision tasks (Davoudi & Komeili, 2021; Donnelly et al., 2022). These prior works establish the value of prototype-based neural nets, but they focus on traditional classification tasks; we are similarly inspired by such methods but seek to build interpretable agents in RL settings.

Whilst we are not the first to build interpretable RL models (Vouros, 2022; Milani et al., 2022), almost all prior research uses interpretable proxy models (such as trees) to imitate agents in symbolic domains. However, these techniques do not apply to richer domains with high-dimensional inputs (such as raw pixels) that we focus on in this work. To date, most work in these deep RL settings has focused on post-hoc approximations involving attention weights (Zambaldi et al., 2018; Mott et al., 2019), or trees (Liu et al., 2018), but these methods do not allow transparency of the agent's actions or intent (Rudin et al., 2022). Another interesting approach distills recurrent neural network (RNN) policies into finite-state-machines (Danesh et al., 2021; Koul et al., 2018), but the approach does not always reveal easily analyzable results, and is restricted to RNNs. Perhaps the most relevant approach is that by Annasamy & Sycara (2019) who learn exemplars to explain Atari games, but their method only works in discrete action spaces, and suffers a performance gap relative to black-box counterparts. In contrast to these approaches, PW-Nets are designed to be human-interpretable, generalize to any neural-based agent with any action space, do not lose model performance relative to black-box agents, and do not rely on post-hoc approximations prone to error.

User studies in *deep* RL trying to predict an agent's actions have shown mixed results (Anderson et al., 2020), with the focus now moving towards arguable more useful tasks such as identifying defective models (Olson et al., 2021). In the prototype literature, studies have generally focused on how "similar" test instances look to prototypes used (Das et al., 2020; Rymarczyk et al., 2021), but this does not evaluate if the explanation is useful in downstream applications, or appropriately calibrates trust in users (Sanneman & Shah, 2022). In contrast to these studies, we ask users to simulate model behaviour and predict failure (and success) cases for the agent, a useful application that should illustrate if trust is appropriately calibrated in the agent's abilities.

## 3 METHOD

Section 3.1 first details our assumptions of a Markov environment and trained neural agent, before Section 3.2 describes the proposed Prototype Wrapper Network (PW-Net), which creates a "wrapper" around the agent to make a *new* end-to-end interpretable policy that reasons with human-defined prototypes. Finally, Section 3.3 gives certain performance guarantees for the system which shows it can always closely approximate the performance of a black-box agent.

### 3.1 MARKOV FRAMEWORK

Our technique assumes access to a neural agent pretrained in a Markov Decision Process (MDP). An MDP is defined by the $(S, A, T, R, \gamma)$ tuple (Sutton & Barto, 2018). $S$ is the set of states; $A \in R^M$ is the sets of $M-$dimensional actions; $T : S \times A \longrightarrow S$ is the probabilistic transition between states due to actions. We note that actions in RL, unlike outputs of simple classifiers, may be multi-dimensional (e.g., a car must control both steering and acceleration at the same time). We therefore dub each dimension in $R^M$ a separate "action". Lastly, $\gamma$ and $R$ define the discount factor and reward function, respectively. In standard RL training methods, the goal is to find the policy, $\pi : s \in S \longrightarrow A$, that maximizes the expected discounted reward. Our approach only requires access to a pre-trained black box policy, $\pi_{bb}$; any existing method from prior art may be used to generate such a policy (Sutton et al., 2000; Williams, 1992). Assuming a neural net instantiation of this policy with a final linear layer, we may decompose $\pi_{bb}$ into an encoder $f_{enc}$, alongside the last layer with weights $\boldsymbol{W}$ and bias $\boldsymbol{b}$ as follows: $\pi_{bb}(s) = \boldsymbol{W} f_{\text{enc}}(s) + \boldsymbol{b}$.

### 3.2 PROTOTYPE-WRAPPER NETWORK

Our primary contribution is a prototype wrapper neural net model, PW-Net, that converts black-box neural models into prototype-based agents for RL by forcing them to use human-understandable

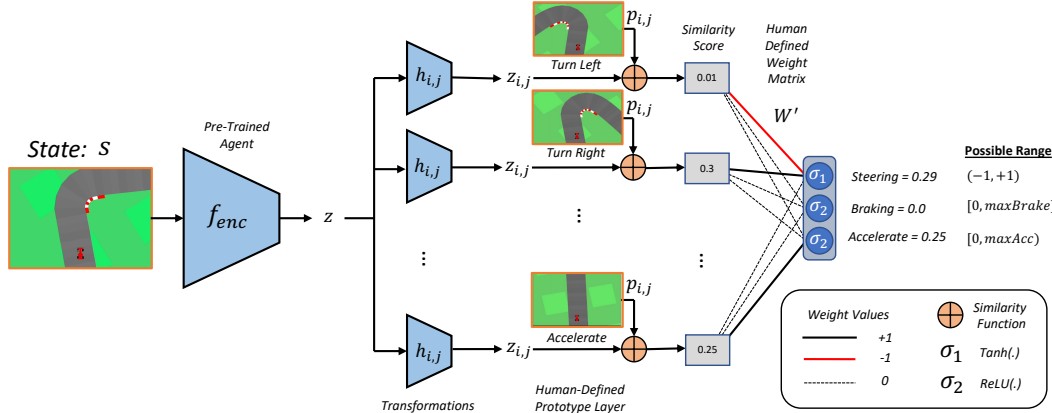

Figure 1: Prototype-Wrapper Network: The framework is instantiated here in the Car Racing domain from OpenAI's gym. A state is encoded as $z$ using $f_{enc}$, before undergoing multiple *separate* transformations and having its similarity to pre-defined human prototypes measured. Each output action can have one or more prototypes associated with it representing some understandable concept(s), all of which contribute to the output. For example, the action "Steering" sums separate prototypes for turning left and right to get a final value, whilst the action "Accelerate" uses only one. The $i, j$ indices are generic placeholders and would be $z_{1,1}, z_{1,2}$... in the example here for "Turn Left" and "Turn Right", respectively. For simplicity and clarity, we ignore the Tanh($\cdot$) function when calculating the "Steering" output in the figure.

concepts, thus creating a new policy. The overall network diagram is depicted in Figure 1, instantiated for a car racing domain in which a simulated car must steer, brake, and accelerate.

In the wrapper model, a state, $s$, is mapped to a latent representation, $z$, via the original encoder network: $z = f_{enc}(s)$. The network defines, for each action $i \in [1, M]$, a set of prototypes that, when combined, dictate action $a[i]$. In Figure 1, for example, the steering action is associated with prototypes for a left and right turn. For each prototype, for each action, a separate "projection network", $h_{i,j}$ maps from $z$ to a specific representation, $z_{i,j}$ for that action and prototype. Within that space for action $i$ and prototype $j$, we use the similarity function from Chen et al. (2019) to score the distance between $z_{i,j}$ and prototype $p_{i,j}$; lastly, those similarity scores are combined via a linear weight matrix, $W'$ to produce actions. This process is defined more formally in Equation 1.

$$z = f_{enc}(s)$$
$$z_{i,j} = h_{i,j}(z) \qquad \forall i \in [1, M], j \in [1, N_i]$$
$$\texttt{sim}(z_{i,j}, p_{i,j}) = \log \left( \frac{(z_{i,j} - p_{i,j})^2 + 1}{(z_{i,j} - p_{i,j})^2 + \epsilon} \right) \tag{1}$$
$$a'_i = \sum_{j=1}^{N_i} W'_{i,j} \texttt{sim}(z_{i,j}, p_{i,j}).$$

We created "human-friendly" prototypes by specifying a set of prototypical states ($s_{pi,j}$) that, when encoded, were used as the latent prototypes $p_{i,j} = h_{i,j}(f_{enc}(s_{pi,j}))$ in the network. For example, in the self-driving car experiments here, one prototypical state was an example of a car turning left and another state was an example of the car turning right; when encoded, these latent prototypes represent different ways of turning. Prior literature in prototype-based neural networks indicates how one could train our wrapper model to learn its own prototypes (Chen et al., 2019). However, our human-defined prototypes are motivated by recent literature showing prototype networks benefit from putting humans in the learning loop (Bontempelli et al., 2023), coinciding with work showing learning human-interpretable disentangled representations is both empirically and theoretically impossible without useful inductive biases or human supervision (Locatello et al., 2019).

We also added additional human supervision by manually defining $W'$ to control the relationship between prototypes and actions. Intuitively, $W'_{i,j}$ corresponds to how prototype $p_{i,j}$ should effect action $a_i$. In our self-driving car example, $a_1$ was the turning rate: by setting $W'_{1,1} = -1; W'_{1,2} =$

---

**Algorithm 1:** Training a Prototype-Wrapper Network

---

**Input:** $f$: A pre-trained agent with encoder $f_{enc}$ and policy $\pi_{bb}$ to "wrap".
**Output:** A PW-Net.

**1** Initialize wrapper, where $W'$ is a manually specified weight-matrix.
**2** Sample $n$ state-action pairs from environment for Dataset $\mathcal{D} \leftarrow \{(s, \pi_{bb}(s)) \sim P(\mathcal{S})\}_{i=0}^{n}$.
**3** Choose Human-Interpretable Prototypical States $\mathcal{S}_p \in \mathcal{S}$.
**4** **foreach** batch $(s, a)$ in $\mathcal{D}$ **do**
**5** $\quad$ $z \leftarrow f_{\text{enc}}(s)$
**6** $\quad$ **foreach** $i$ in `range`$(|M|)$ **do**
**7** $\quad\quad$ $a'[i] \leftarrow 0$
**8** $\quad\quad$ **foreach** $j$ in `range`$(|S_{pi}|)$ **do**
**9** $\quad\quad\quad$ $p_{i,j} \leftarrow h_{i,j}(f_{\text{enc}}(S_{pi,j})$
**10** $\quad\quad\quad$ $z_{i,j} \leftarrow h_{i,j}(z)$
**11** $\quad\quad\quad$ $a'[i] \leftarrow a'[i] + W'_{i,j} \, \text{sim}\, (p_{i,j}, z_{i,j})$
**12** $\quad$ Minimize Loss $\mathcal{L}(a'|a; f_{enc}; \phi; W')$ with gradient descent, updating only $\phi$.
**13** Cache all $p_{i,j} = h_{i,j}(f_{\text{enc}}(s_{pi,j}))$ for testing time inference.
**14** **return** Trained PW-Net

---

$+1$, we specified that the car's turning rate should equal the difference between the similarity to a prototypical left and right turn. In principle, one could combine many similarity scores for the same action, but from a performance perspective, we never found the need to use more than two (although this is likely situation- and task-dependent). $W'$ guarantees that each prototype is associated with an intuitive action(s), if it was learned then each prototype could correspond to multiple actions which may be undesirable. For instance, keeping with the car example, we don't want the prototype image for "Accelerate" to contribute to the output action "Braking" (see Fig. 1).

Given manually-specified prototypical states and $W'$, the only trainable parts of PW-Net were the projection networks, $h_{i,j}$, with parameters $\phi$. These multiple transformations were necessary for optimal performance (as opposed to putting all prototypes in a single latent space). In experiments, we trained $h_{i,j}$ using gradient-based optimization to minimize a supervision loss between the PW-Net and the black-box policy $\pi_{bb}$ in specific states, whilst also constraining it to use the prototypes. The training data for this process was generated by running the agent in its native domain for $n$ iterations and collating the state action pairs into dataset $\mathcal{D} \leftarrow \{(s, \pi_{bb}(s)) \sim P(\mathcal{S})\}_{i=0}^{n}$.

PW-Net's explanation is a classic case-based reasoning one, where explanations are of the form *"I am going to do action x, because this situation is similar to a previous prototypical situation when I also did action x"* (Leake, 1996; Keane & Kenny, 2019). As our model is inherently interpretable, the explanation does not need to be generated, it is naturally produced when deciding each action to take. The explanation can be "local", explaining individual actions of an agent (see Fig. 1), or "global", explaining the car's understood concepts, which we use for our user study (see Fig. 3).

### 3.3 PERFORMANCE GUARANTEES

Here, we show that PW-Nets can learn a comparable policy to the black-box model, indicating that performance is unlikely to degrade when using our more interpretable architecture. Given that both the black-box model and PW-Net share a common encoder, $f_{\text{enc}}$, we therefore seek to prove that the output of a PW-Net from $z$, which we dub $f'_{pw}(z)$, can equal the output of the black-box model from $z$, $f_{bb}(z)$, for all $z$. For simplicity, we consider only the first action, $a_1$ with:

$$
\begin{aligned}
a'_1 &= \sum_{j=1}^{N_i} W'_{1j} \, \text{sim}(z_{1j}, p_{1j}) \\
&= W'_{11} \log \left( \frac{(h_{11}(z) - p_{11})^2 + 1}{(h_{11}(z) - p_{11})^2 + \epsilon} \right) + W'_{12} \log \left( \frac{(h_{12}(z) - p_{12})^2 + 1}{(h_{12}(z) - p_{12})^2 + \epsilon} \right) \\
&= W'_{11} \log \left( \frac{d_{11}^2 + 1}{d_{11}^2 + \epsilon} \right) + W'_{12} \log \left( \frac{d_{12}^2 + 1}{d_{12}^2 + \epsilon} \right).
\end{aligned}
\tag{2}
$$

Equation 2 simply restates how action $a_1$ is calculated based on prototypes. For simplicity, we only consider two prototypes per action (thus, two entries for $W'_{1,j}$). The range of the similarity function is $(0, \log(1/\epsilon)]$; thus, the range of possible values of $a'_1$ is set by $W'_{1j}$. Assuming $W'_{1,1} > 0; W'_{1,2} < 0, a'_1 \in [W'_{1,2} \log(1/\epsilon), W'_{1,1} \log(1/\epsilon)]$. We note that the similarity function decreases monotonically with distance, so it is invertible for legal values. Therefore, even if $p_{i,j}$ were fixed, a sufficiently-parameterized $h_{i,j}$ could learn to output any similarity score simply by placing $z_{i,j}$ at the correct distance from $p_{i,j}$. Hence, as long as the black box policy outputs always fall within a finite range, PW-Net can closely match the output. In experiments, we found one linear layer was not enough to accomplish this, two was sufficient (which we used), and more showed no benefit.

Two important assumptions are worth highlighting. First, we assumed that for each action there were at least two entries in $W'$ and that they had opposite signs. The similarity function always generates positive outputs; thus, to match a black box policy with both positive and negative values, it is important to use both positive and negative weights in $W'$. In practice, we found this necessary in the self-driving car experiment. Second, we assumed independent $h_{i,j}$ for all $i, j$. Prior work in prototype-based neural nets often locate all prototypes within a single latent space. However, given a fixed $W'$, and without separate $h_{i,j}$, every state-action pair generated by $\pi_{bb}$ creates a set of (potentially incompatible) constraints for the relative locations of prototypes. In practice, we found that learning separate $h_{i,j}$ was necessary for performance, which we show in the next two sections.

The earlier discussion found that PW-Nets may mimic a black-box policy for almost all $z$ by assuming that $h_{i,j}$ may be learned to output any desired similarity score between $z_{i,j}$ and $p_{i,j}$. However, there are a small number of notable exceptions: precisely the $z$ corresponding to prototype states. By definition, the distance between between such encodings and the prototype will be zero, which specifies the PW-Net output for such states. Recall however, we are not trying to perfectly mimic the agent, instead we seek to use it as a "guide" whilst learning a new policy that reasons with the prototypes. In doing so no notable performance degradation was noticed, in fact, we often observed performance improvements by forcing the network to utilize the human-defined prototypes.

## 4 CONTINUOUS ACTION SPACES

Here we test if PW-Nets can perform on par with any black-box agent by testing them in a deep RL domain with multiple continuous output actions and a state space of only raw pixels. In addition, another domain with symbolic inputs is tested to further display generality. PW-Nets are compared against several baselines and the best methods are determined by the average reward. As a sanity check, we also report how closely the methods imitate the black-box predictions at test time.

### 4.1 EXPERIMENT SETUP

The deep RL domain tested here is the Car Racing problem from OpenAI's gym environment, the agent receives rewards for driving around the track as fast as possible and avoiding swerving off the road. We use a pre-trained model (partly to avoid any bias) from Jain (2022) which solved the environment using Proximal Policy Optimization (PPO). The agent has three continuous output actions (see Fig. 1). Similarly, for a symbolic problem, the Bipedal Walker environment from Open AI's gym Box2d was chosen, where the agent receives rewards for simply walking straight and not falling over. For this latter domain a pre-trained model from Barhate (2022b) was used which solved the problem with the Twin Delayed DDPG (TD3) algorithm. The agent has four continuous output actions (see Appendix A for prototypes used).

We compared first to $k$-Means in the space $z$ with the same number of clusters as the number of prototypes used in PW-Net, before mapping each centroid onto the nearest training example to get the prototypes. Then, in contrast to the other methods, we allowed $k$-Means to learn the weight parameters in the last layer to help it. As another baseline, we followed the same protocol as PW-Net, but only a single transformation is used which houses all prototypes together in the same latent space which we call PW-Net*. In addition, this baseline learns the prototypes as trainable parameters, and once training is complete, they are mapped onto the nearest training example. This can be best thought of as an ablation test of PW-Net, however it bears resemblance to Chen et al. (2019) and their ProtoPNet model, so we also implemented the clustering and separation losses suggested by the authors (adapted for continuous domains) to help the baseline further (see Appendix B). Lastly,

Table 1: Continuous Action Spaces: Various baselines are compared to PW-Net. Overall, only PW-Net is comparable to the performance of the original black-box agent in both domains.

| Method | Car Racing | | Bipedal Walker | |
|---|---|---|---|---|
| | Reward | MSE | Reward | MSE |
| PW-Net | **220.72 ± 0.34** | **1.37 ± 0.16** | **304.77 ± 6.72** | **0.65 ± 0.38** |
| VIPER | N/A | N/A | -89.71 ± 7.51 | 101.15 ± 8.16 |
| PW-Net* | -9.48 ± -02.50 | 16.93 ± 3.01 | 190.41 ± 59.51 | 69.77 ± 15.16 |
| k-Means | -2.09 ± -00.94 | 29.04 ± 0.01 | -107.72 ± 0.13 | 136.90 ± 0.72 |
| Black-Box | 218.41 ± 03.00 | N/A | 310.71 ± 1.47 | N/A |

we also compare to VIPER by Bastani et al. (2018), but we adapt their algorithm here by using uniform sampling (which is implemented in their code base as an alternative) instead of the weighted sampling they discuss (which needs q-values), which allows us to use it in experiments.

Two metrics were used, average reward and mean-squared error (MSE). Reward was calculated by training each method 5 times and taking the average reward accumulated across 30 simulations, these 5 means then produced a final mean and standard error. The second metric took the MSE of the black-box model's actions compared to the interpretable model's actions each iteration, to see how closely methods approximate the oracle. As PW-Net is optimizing a function which (1) matches the original agent, and (2) uses the human-defined prototypes, it will not 100% match the agent's predictions, and indeed we don't want it to, we want it to learn a new policy which reasons with the prototypes and $W'$. MSE is included to make sure part (1) of the optimization is working.

### 4.2 RESULTS

Table 1 shows the results in continuous action domains. In the deep RL domain Car Racing, only PW-Net (M=220.72, SE=0.34) was capable of achieving a similar average reward as the black-box (M=218.41, SE=3.). Moreover, PW-Net achieved the closest predictions to the black-box (i.e., the MSE was smallest). In Bipedal Walker, again PW-Net (M=304.77, SE=6.72) was capable of achieving near the same average reward as the black-box (M=310.71, SE=1.47), although PW-Net* did achieve notable reward too (M=190.41, SE=59.51). $k$-Means performed the worst overall, with the lowest reward, with VIPER fairing only slightly better. Note that VIPER only works in symbolic domains, so we could not test it on Car Racing.

## 5 DISCRETE ACTION SPACES

Here again PW-Nets are tested, but in discrete domains to help prove generality. As before, a deep RL domain using raw-pixels as state representations is tested, alongside a symbolic domain. The same baselines and metrics are used, although with some minor modifications.

### 5.1 EXPERIMENT SETUP

The deep RL domain tested is the deterministic version of Atari Pong from OpenAI's gym environment, the agent receives rewards for scoring against the opponent, and losses rewards for being scored against. The agent uses a state space of raw pixels, has six discrete actions corresponding to moving up or down, and we use a pre-trained model from Şentürk (2022) which solved the environment using the DQN algorithm (and is 5% stochastic). For a symbolic problem, the Lunar Lander environment was used, where the agent gets reward by landing the spaceship successfully in the shortest time possible. The agent has four actions corresponding to doing nothing, or firing its engine on three different sides. A pre-trained model from Barhate (2022a) was used which solved the problem with the TD3 algorithm (see Appendix A for prototypes).

As baselines, we again compared to $k$-Means, but this time by dividing the training data in the space $z$ using each separate action, and clustering each with $k = 1$, before mapping each centroid onto the nearest training example. Then, in contrast to the last experiment, we didn't allow $k$-Means to

Table 2: Discrete Action Spaces: Various baselines are compared. However, again, only PW-Net consistently performs well.

| Method | Atari Pong | | | Lunar Lander | |
| | Reward | Accuracy | | Reward | Accuracy |
| --- | --- | --- | --- | --- | --- |
| PW-Net | **10.72 ± 0.26** | **88.93 ± 0.00** | | **216.94 ± 16.92** | **97.63 ± 0.00** |
| VIPER | N/A | N/A | | -408.81 ± 60.98 | 59.26 ± 1.01 |
| PW-Net* | 8.85 ± 1.69 | 84.84 ± 0.76 | | 124.54 ± 120.53 | 88.67 ± 0.01 |
| k-means | -21.00 ± 0.00 | 11.79 ± 4.15 | | -419.46 ± 119.08 | 10.10 ± 5.87 |
| Black-Box | 11.94 ± 0.16 | N/A | | 212.94 ± 2.63 | N/A |

learn the weight parameters in the last layer since each can easily be associated with an appropriate action. PW-Net* was again used here, but this time using the exact same clustering and separation losses suggested by Chen et al. (2019), since they developed the losses for discrete classification problems they will work here. Lastly, we again compare to VIPER by Bastani et al. (2018) using the uniform sampling variation of their algorithm.

Two metrics were used, reward and accuracy. Reward was calculated the same way as in our previous experiment. The second metric took the accuracy of the black-box model's actions compared to the interpretable model's actions at test time. This has the same functionality as MSE in the previous section, but now adapted for discrete domains.

## 5.2 RESULTS

Table 2 shows the results of using PW-Nets in discrete action spaces. In Pong, PW-Net achieved the closest reward (M=10.72, SE=0.26) to the black-box's reward (M=11.94, SE=0.16). Interesting divergences happen where PW-Net* achieves lower Accuracy (M=84.84, SE=0.76) compared to PW-Net (M=88.93, SE=0.00), indicating that PW-Net was better at learning from the black-box. In Lunar Lander, again only PW-Net variations were able to reach a comparable reward to the black box. Specifically, PW-Net achieved a reward (M=216.94, SE=16.92) which was comparable to the black-box (M=212.94, SE=2.63), with PW-Net* doing notably worse (M=124.54, SE=120.53). Larger divergence happens when comparing accuracy, as PW-Net (M=97.63, SE=.0) achieves significantly higher results than PW-Net* (M=88.67, SE=0.01), indicating that the former is again better able to mimic the original black-box function. Although marginally better than $k$-Means, VIPER performs badly overall (M=-408.81, SE=60.98), perhaps suggesting that decision tree proxy models will struggle in their current form to generalize beyond relatively simple domains such as Cartpole where they are usually tested successfully (Bastani et al., 2018; Silva et al., 2020). Lastly, note the Pong model was 5% stochastic, so reaching high performance with PW-Net was impossible, as it was trained on 5% random data, and tested with an additional 5% stochasticity on top of this.

So, considering the two experiments thus far, it is clear that PW-Net generalizes the best overall both in terms of reward and MSE/Accuracy, particularly in deep RL settings which is our focus here. A natural question to ask is how sensitive the performance of PW-Net is to the prototypes chosen, so we performed a test on all four domains where the prototypes were randomly chosen from the training data 30 times with 30 simulations ran each time. Results showed no notable difference to the black-box in Car Racing (Reward: M=218.57, SE=0.09, MSE), Atari Pong (Reward; M=10.71, SE=0.29), Lunar Lander (Reward; M=213.78, SE=11.19), or Bipedal Walker (Reward: M=309.44, SE=1.79). So, overall, PW-Nets are very robust to the prototypes chosen. However, as we shall now see, user task performance crucially depends upon choosing human-friendly prototypes.

## 6 USER STUDY

Here we test if PW-Nets can improve a user's ability to predict in distribution (ID) successes and out-of-distribution (OOD) failures in a simple self-driving car domain, an application seen as highly sought after (Rudin et al., 2022). We setup the materials such that all ID situations were driven successfully by the car, and all OOD situations were ones in which it crashed (see Fig. 5). The

explanation shown to users is a global explanation of the car, which for the experiment group using PW-Net is shown in Figure 3.

Two different controls were compared to our "Human-Friendly Prototype" experiment group which used PW-Net's global explanation. The first is a "Non-Human Friendly" group which uses the same prototypes as our network, but rearranges them randomly. So, the weight matrix and actions remain the same, but different prototype images are associated with each. The purpose of this comparison is to show that although the performance of the network itself does not hugely hinge on associating human-friendly prototypes with intuitive actions, the performance of humans does. Note this is an actual PW-Net we trained that performed just as well as the other models here (even though the prototypes were purposefully "badly chosen"). This group was easily controlled because they saw the same amount of information in each question as the experiment group, whilst being presented in a similar way (see Fig. 6 and 7). Secondly, we compare our network to a full black-box, where people are simply told a "no explanation explanation" in text form with *"The car has learned to drive very well by practicing for millions upon millions of simulations"*. This group was controlled by replacing the prototype images with this text (see Fig. 6). The rationale here is that these images needed to be replaced with comparable information in order to control the cognitive load between groups. After seeing the current situation of the car, participants are shown the explanation, and then asked to predict if the car will drive safely or not on a scale of 1-5, where 1 corresponds to "Strongly Disagree" and 5 to "Strongly Agree". Note numeric scales were used (not a traditional Likert one) so the data could be confidently analyzed using t-tests (Carifio & Perla, 2008).

So, the two hypotheses tested are: **H1:** Users require human-interpretable prototypes to perform well in model simulation tasks. **H2:** When people receive "no explanation", they will have inappropriately high trust.

**Participants.** Comparing to similar studies (Kenny et al., 2021), we anticipated a medium effect size, but pilot studies showed a large effect. Hence, anticipating a medium-large effect, we chose an alpha of 0.05, a power of 0.8, and an effect size of 0.65, which predicted through a power analysis that a sample size of 30 was required in each group to avoid $p$-hacking. Hence, 90 people in total were recruited from prolific.co, 30 per group, all paid $12 per hour to complete the study. All participants were restricted to being native English speakers, using desktop computers, in either the U.S. or the U.K. This study passed ethics review of the institution.

**Materials.** 20 materials were randomly shown to users, 10 in distribution (ID), and 10 out-of-distribution (OOD), the latter was generated from the open source modification to the Car Racing environment provided by Woodcock (2016), 5 of which showed red obstacles the car should avoid, and the other 5 of which showed unusual roads not seen normally in the environment (such as t-junctions). All ID situations were gathered form the original domain, driven successfully by the car, and labelled as "driving safe". The OOD situations were real failure cases of the car, where all trained models either (1) drove off the road crashing, or (2) hit the obstacles. If the car did either of these things, it was labelled as "not driving safely". All these questions can be seen in Figure 5 to make the ID and OOD materials clear. After the questions, users where shown where the car did/didn't drive safe, and asked to complete the NASA TLX perceived workload questionnaire.[1]

## 6.1 RESULTS.

Each participant's scores were averaged into two numbers for both their ID and OOD materials, which formed the distributions of interest. **H1:** Fig. 2(A) shows a large difference in predicting ID success. Specifically, those who received "Human-Friendly Prototypes" (M=4.48, SD=0.43) compared to participants in the "Non-Human Friendly Prototypes" control group (M=2.19, SD=0.78) demonstrated better ability to predict when the system would drive safe, $t(58)=13.8$, $p < .0001$, showing human-friendly prototypes had a large effect on people's task performance. This lends evidence that prototypes need to be carefully chosen in order to maximize the effectiveness of PW-Net global explanations. **H2:** Fig. 2(B) shows a significant difference in predicting OOD failures between participants who received "Human-Friendly Prototypes" (M=2.24, SD=0.68) and participants in the "Black-Box" control group (M=3.48, SD=0.75). The prior demonstrated significantly better

---

[1]Users were also asked for their confidence in predictions, and an additional system usability questionnaire was asked. However, no results were significant, so they are not reported.

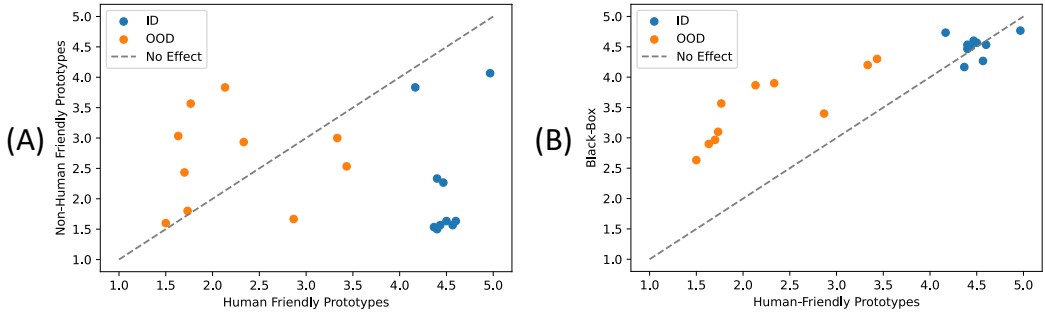

Figure 2: User Study Results: Ratings for how confident users were the car would drive safely (5 rating) v. not drive safely (1 rating) for in distribution (ID) v. out-of-distribution (OOD) data. Each point on the plots represents the average of a single question in the survey. Compared to the experimental group, *Non-Human-Friendly* control group were significantly worse at predicting when the car would drive safely, and the *Black-Box* group at predicting when it would crash. The plots can be interpreted as datapoints close to the dashed line showing no difference between groups, whilst points far from it showing significant differences.

ability to predict when the system would *not* drive safe, $t(58)$ = -6.53, $p < .0001$, showing that prototypes give a global understanding of the system which allows people to predict OOD-based system failures better than no explanation. This is interesting because it shows that people who see no explanation are overly confident in the black-box's abilities, leading to inappropriately high trust, which PW-Nets actively avoid by appropriately lowering their trust here.

Aside from testing our original hypotheses, additional post-hoc analysis considered the groups from a pure accuracy perspective, where scores of 4/5 were taken as the user predicting the car would drive safely, and scores of 1/2 as the user predicting the car would not drive safely, a score of 3 was always marked as incorrect, since the user is uncertain either way. The human-friendly group was better at predicting if the car would drive safe or not (M=0.75, SD=0.14), than the non-human friendly group (M=0.37, SD=0.15), and the black-box group (M=0.57, SD=0.10), with both results achieving statistical significance $p < .05$. This ability to accurately predict a system's capabilities is considered an important proxy metric for trust calibration (Hoffman et al., 2018; Sanneman & Shah, 2022). Hence, it is reasonable to posit that PW-Nets may have a role to play in calibrating appropriate user trust of AI systems, something which is seen as a core application for XAI, but has proven quite difficult to demonstrate in practice (Silva et al., 2022; Miller, 2022).

Lastly, the NASA TLX questionnaire showed there was a significant difference in people's perceived workload between participants who received "Human-Friendly Prototypes" (M = 5.80, SD = 1.74) and participants in the "Non-Human Friendly" control group (M = 7.25, SD = 2.63). The control recorded a higher workload, $t(58)$ = -2.45, $p < .02$, showing that badly chosen prototypes make the task of the using the system significantly more taxing on people.

## 7 CONCLUSION AND FUTURE WORK

The world has grown increasingly excited about possible applications for deep RL. However, without transparency in the agent's actions and intent, it is not feasible to use the systems in truly sensitive domains (Rudin, 2019). Here, we showcased an accessible way to force any deep RL agent to use human understandable concepts in an interpretable way with example-based reasoning (Kenny & Keane, 2021). Results demonstrated that the technique does not lose model performance and generalizes across discrete and continuous action spaces. In addition, a user study showed real applicability of the technique to assist human task performance in predicting model failure (and success) states, hence calibrating *appropriate* user trust. In future work, it would be interesting to experiment with learning the prototypes and/or weight matrix, particularly in domains people cannot easily define concepts. In addition, applying PW-Nets to supervised/self-supervised learning, fine-tuning (or training from scratch) the pre-trained encoder, incorporating feature saliency, exploring prototypical trajectories, and adding additional interpretability constraints to the loss could all be explored.

ACKNOWLEDGEMENTS

This research was supported in part by Army Research Lab grant #1130252-431282. The authors would like to thank the anonymous ICLR reviewers who greatly helped to improve the presentation of this work. In addition, we are grateful to our lab members who supported this work with some early advice and helpful critiques.

REPRODUCIBILITY STATEMENT

The code to reproduce the results is available at:

`https://github.com/EoinKenny/Prototype-Wrapper-Network-ICLR23`

For the user study, we also included all four group surveys as PDFs in the repo, so people may recreate the study results if they wish. If researchers are interested in reproducing the survey results more easily, please contact the corresponding author and we will share the Qualtrics surveys.

ETHICS STATEMENT

One of the potential ethical considerations of using prototypes for explanations is that the examples used must perserve user privacy. Considering this, if PW-Nets were used in downstream applications, some consideration is going to need to be given to this. However, when compared to more generic nearest neighbor algorithms, PW-Nets have a big advantage here in that there is a much smaller group of exemplars which may be possibly presented to users, so the efforts on algorithm designers to preserve privacy will be much easier.

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

## A    PROTOTYPES CHOSEN

In this appendix we explain how the prototypes were chosen for each domain.

### A.1    CAR RACING

In this domain there are three action outputs, steering, braking, and accelerating. Steering has a possible range from -1 to +1, with negative values turning left, and positive values turning right, 0 means to drive straight on. Accelerating and Braking both have values from 0 to $\infty$.

When choosing the prototypes for PW-Nets, we broke steering down into two concepts of "turning left" and "turning right", with a prototype representing each concept. For accelerating and braking, we only chose one prototype each. For accelerating a state of a straight road was chosen, since this is typically seen as a safe situation to accelerate, and for braking the motion of taking a turn was chosen, since it is necessary for a car to always brake when turning a corner in order to drive safely, these where seen as being somewhat interpretable for people (see Fig. 3).

Note that many more concepts could be used to dictate when the car brakes or accelerates. For example, if there are obstacles on the road, then one could correspond a prototypical example of these objects to the action "braking". That way, when the car sees such an obstacle, it will brake and explain why is is braking (i.e., due to its similarity to the prototype with the obstacle). However, here for simplicity (and to demonstrate multiple/single options), we only associate one concept with the actions braking and accelerating, whilst associating two with steering.

### A.2    ATARI PONG

In this domain there are six action outputs, action 0 and 1 are placeholders and do nothing, action 2 & 4 makes the racket go up, and action 3 & 5 makes the racket go down. The prototypes used in our tests are shown in Fig. 4, with near identical ones used for the duplicate actions.

### A.3 Symbolic Domains

Although the focus of this paper is on deep RL, these domains were included to help prove the generality of PW-Nets as proposed in Section 3.3. For Lunar Lander and Bipedal Walker we defined "ideal" prototypical examples of each concept in the action-space where the action of interest was 1 or -1 and all others 0, which we then mapped to the nearest training example. Lunar Lander had 1 for the action of interest, and 0 for the others (four prototypes in total). Bipedal Walker had two prototypes for each output action since they ranged from -1 to +1 (so 8 prototypes in total), each representing the concept of one of its limbs moving forward or backwards (e.g., "Right Hip Moving Forwards").

## B  Losses in PW-Net*

When learning all the prototypes in the same latent space $z$, we followed prior literature in Chen et al. (2019) and used their separation and clustering losses as described below:

$$Clst = \frac{1}{n} \sum_{i=1}^{n} \min_{j:p_j \in P_{y_i}} \min_{z \in patches(f(x_i))} \|z - p_j\|_2^2 \tag{3}$$

$$Sep = -\frac{1}{n} \sum_{i=1}^{n} \min_{j:p_j \notin P_{y_i}} \min_{z \in patches(f(x_i))} \|z - p_j\|_2^2 \tag{4}$$

where $n$ is the number of instances in a batch, $j$ is a prototype for one of the classes in $P$, $f(.)$ is the encoder, and $z$ is the latent representation of some patch in test instance $x_i$.

However, in our domains we are considering whole instances, and not parts of them, so the losses become:

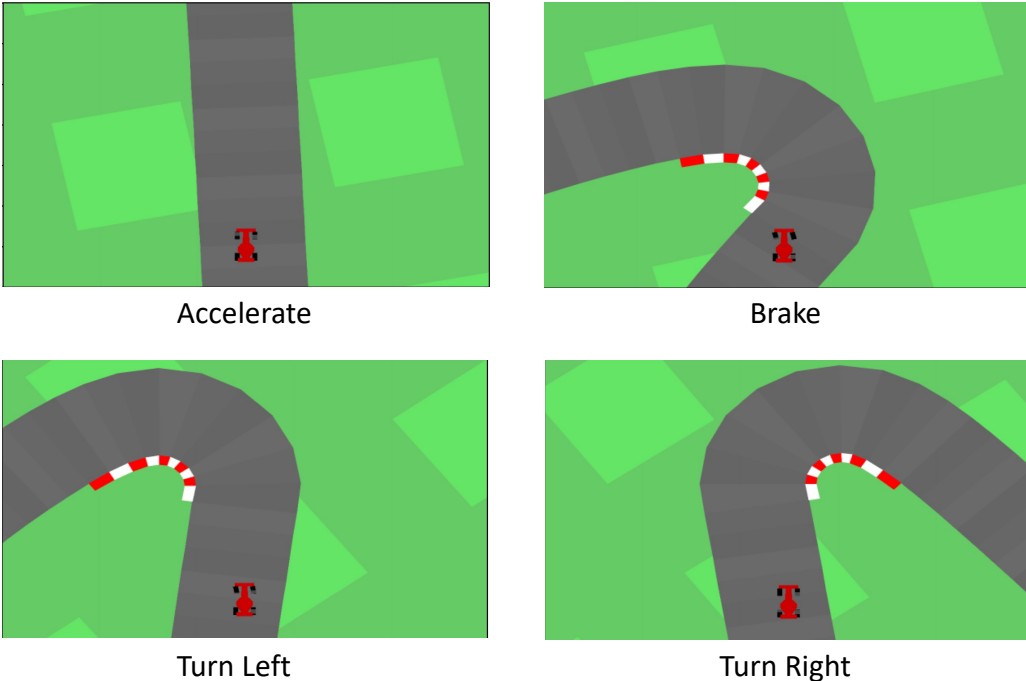

Figure 3: Human-Friendly Prototypes for Car Racing: The four prototypes chosen for Car Racing in our tests. The two prototype for "Turn Left" and "Turn Right" contribute to the output action steering. The two other prototypes singularly contribute to the actions accelerate and brake, respectively.

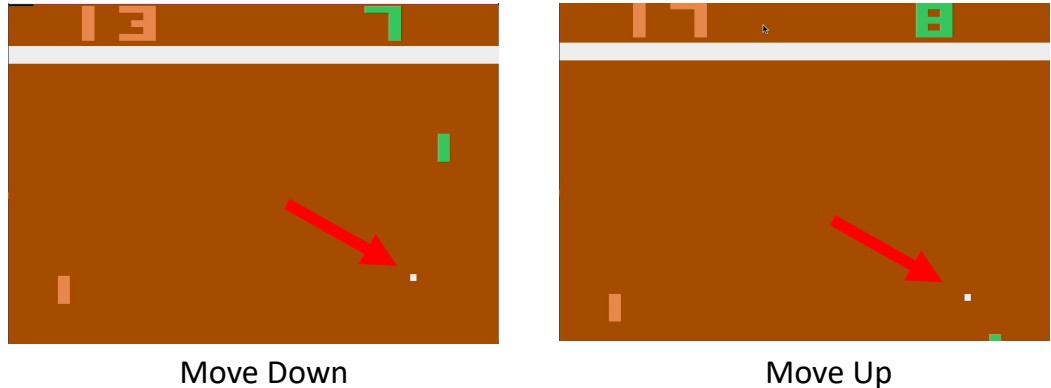

Move Down         Move Up

Figure 4: Human-Friendly Prototypes for Pong: The player is the green object which must hit the white ball back to the left side of the table. The red arrow shows the direction the white ball is travelling.

$$Clst = \frac{1}{n} \sum_{i=1}^{n} \min_{j:p_j \in P_{y_i}} \min_{z=(f(x_i))} \|z - p_j\|_2^2 \tag{5}$$

$$Sep = -\frac{1}{n} \sum_{i=1}^{n} \min_{j:p_j \notin P_{y_i}} \min_{z=(f(x_i))} \|z - p_j\|_2^2 \tag{6}$$

In the continuous actions spaces, the losses had to be modified to work, so they became:

$$Clst = \frac{1}{n} \sum_{i=1}^{n} \sum_{j=1}^{P} \min_{z=(f(x_i))} \|z_i - p_j\|_2^2 \tag{7}$$

$$Sep = -\sum_{i=1}^{P} \sum_{j=1}^{P} \|p_i - p_j\|_2^2 \quad \text{s.t.} \quad i \neq j \tag{8}$$

where $P$ simply represents the number of prototypes used. Intuitively speaking, the first loss forces all prototypes to be close to the data distribution, so that when they are mapped to real instances they do not change much. The second loss forces all prototypes to be far apart from each other. Each loss had to be scaled in optimization with parameters $\lambda_1$ and $\lambda_2$, respectively. In discrete domains, we set the values to 0.8, and 0.08 [following Chen et al. (2019)], but in continuous domains we found 0.08, and 0.008 worked better.

## C  USER STUDY DESIGNS

Rather than include the user surveys in this manuscript, we uploaded them in the accompanying zip file (and repo) in the interests of keeping this document to a reasonable size (both in terms of file size and document length). However, we do show some screenshots here in Appendix E.

## D  TRANSFORMATION ARCHITECTURE

Each PW-Net model here used multiple transformations $h_{i,j}$ to convert $z$ into a latent space in which it could have its similarity measured against human-defined prototypes. The architecture for each of these is specified in Table 3. In practice, the prototype sizes were usually 50 in length, but as little at 5 also worked occasionally.

Table 3: The transformation architecture used in all experiemnts.

| Transformations $h_{i,j}$ | |
| --- | --- |
| Layer | Layer Parameters |
| Linear | (Latent $z$ size, prototype size) |
| InstanceNorm1d | prototype size |
| ReLU | |
| Linear | (prototype size, prototype size) |

## E   USER STUDY: SURVEY DETAILS

Our study was designed to see if people can predict OOD-based failures and ID-based successes based on the explanations given. For the ease of readers, we have included all OOD/ID materials shown to users in Figure 5. Then, Figures 6 and 7 show an example of the actual survey questions asked in all four groups. Please consult the supplementary material or GitHub repo to see the full surveys. Note we are happy to share the surveys so other researchers do not have to rewrite them from scratch if they wish to reproduce the results.

### E.1   THE TRAINING DATA GROUP: OMITTED RESULTS FROM USER STUDY

Note that there was a fourth group in our user study dubbed the "Training Data" group (see Fig. 7B). However, due to concerns the *Training-Data* group here was not as well controlled as the others (due to an additional text prompt reminding users that they are only seeing a small sample of training data), we removed it from the main paper, but we show it here in Figure 8 in case some readers are interested in the results nevertheless. This additional text was added as an attempt to control for the fact users were not reading associated actions with each prototype in the group, so they needed additional cognitive load to be equal to the experimental group. However, it was ultimately decided to omit the group from the main paper as it was seen as too likely to be a possible confounding factor.

Specifically, Fig. 8(A) shows a significant difference in predicting OOD failures. In particular, when comparing participants who received "Human-Friendly Prototypes" (M=2.24, SD=0.68) to participants in the "Training-Data" control group (M=2.79, SD=0.57), the prior demonstrated better ability to predict when the system would *not* drive safe, $t(58)$ = -3.32, $p < .002$. This shows that prototypes give a global understanding of the system's capabilities which allow people to confidently predict OOD-based system failures better than observing segments of training data. However, we reiterate that these results may not reproduce if the instructions in Step 2 were more minimal (see Fig. 7B).

## F   ADDITIONAL PW-NET RESULTS

To help further prove the generality of PW-Nets, we include some more case studies here of their performance in different domains.

### F.1   CART POLE

Here, we trained a DQN model to perform in the cart-pole domain from Open AI's gym environment. It is a symbolic domain with a four-dimensional state space corresponding to cart position, cart velocity, pole angle, and pole velocity at tip. The output actions are simply to go left or go right. To succeed in the environment, the agent must remain balanced for 200 actions.

The DQN learned to perform perfectly in the domain, achieving an average reward of 200 across 150 simulations. Following Algorithm 1, we first collected simulation input-output data for 1000 simulations, corresponding to 20,000 datapoints for training the PW-Net. To pick the prototypes, this data was partitioned into two groups corresponding to each action, and the mean centre of the cluster was used to get the prototype for each action which was the closest mapping in the real data.

In Distribution Materials (ID)

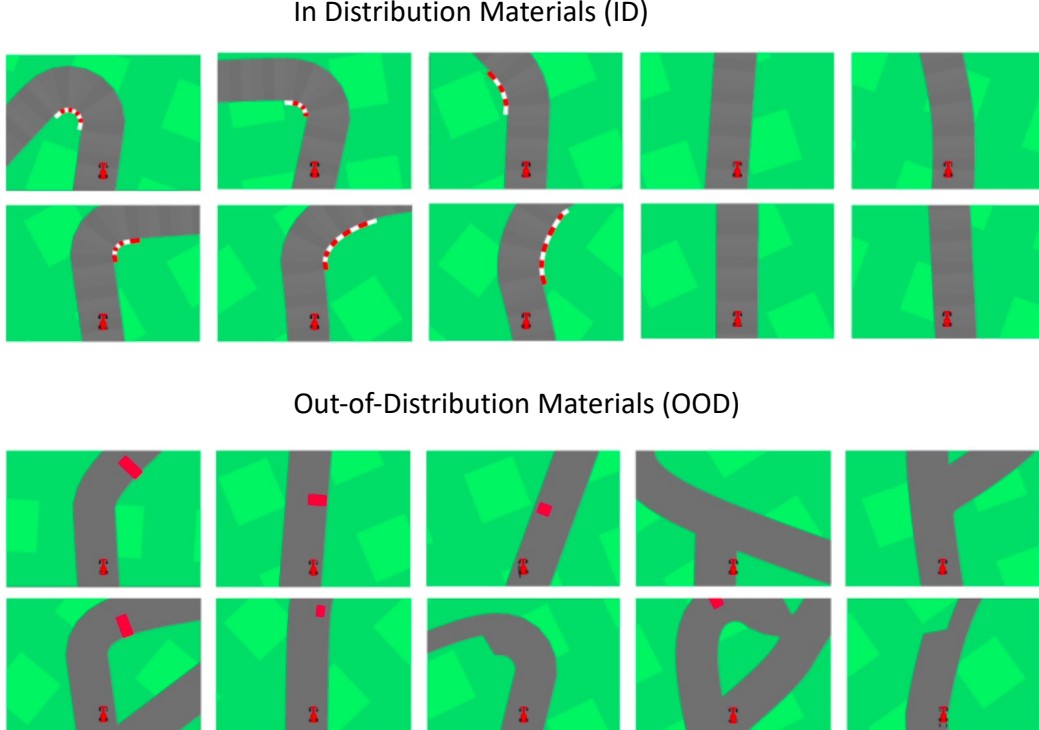

Out-of-Distribution Materials (OOD)

Figure 5: User Study Materials: The ID and OOD materials shown to users in the study. The ID materials were gotten from the domain the model was trained in, whilst the OOD materials were gotten from Woodcock (2016).

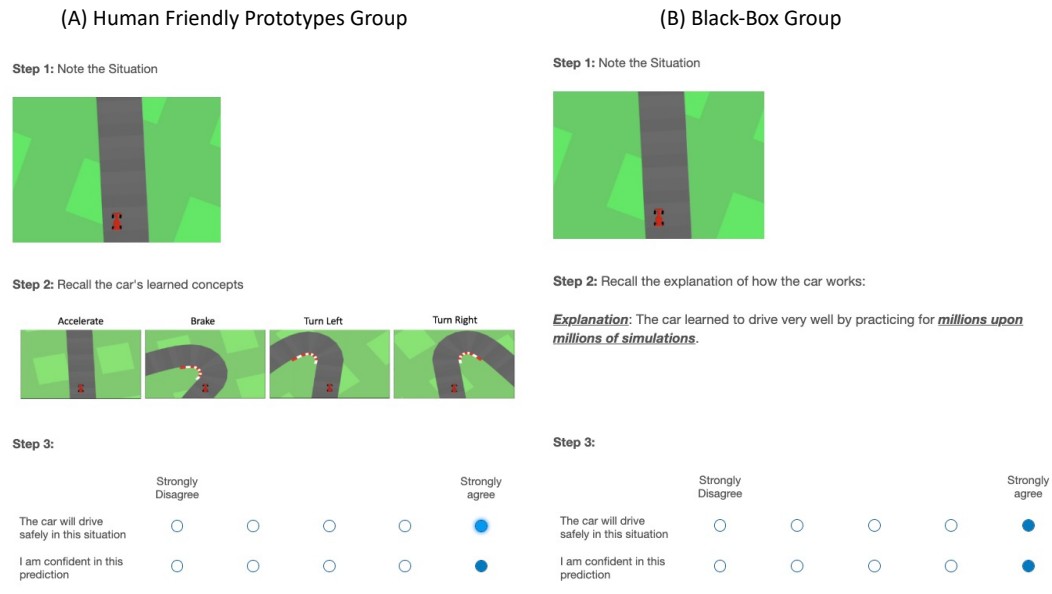

Figure 6: Example Question: (A) The Human-Friendly Prototype group, and (B) the Black-Box group.

We trained five PW-Nets on these data, for 30 epochs, and ran 30 simulations each time to collect the results. Overall, the PW-Net perfectly performed in the domain with an average reward of 200

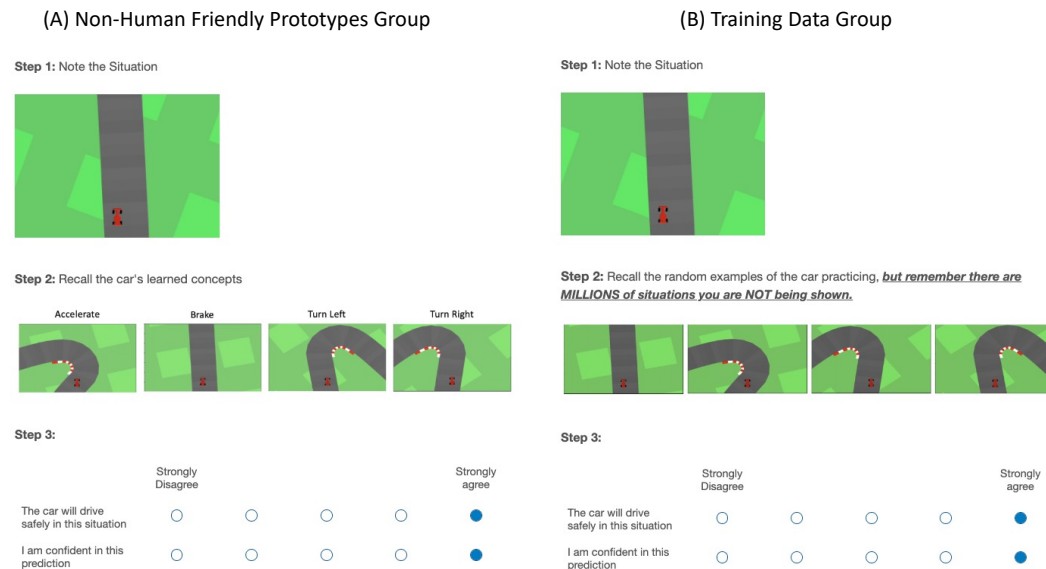

Figure 7: Example Question: (A) The Non-Human-Friendly Prototype group, and (B) the Training-Data group.

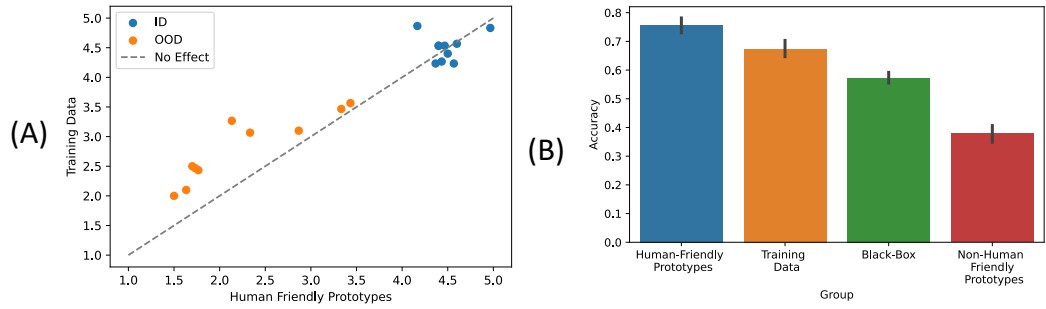

Figure 8: Additional User Study Results: (A) The "Training Data" group from our user study shows how people are less able to confidently predict OOD errors where the car crashes into red obstacles or runs off the road. (B) The accuracy of all groups are plotted in a bar chart showing their ability to predict if the car will crash or not. Standard Error bars are shown in (B). The results of (A) were omitted from the main study due to some concerns the extra text in Step 2 of Figure 7 confounded the results.

$\pm$ 0.00, and an accuracy of 98.81%, showing that it approaches the same function as the black-box, but the constraints of using the prototypes means it inevitably learns to reason in a different manner more intuitive for humans.

The other algorithms PW-Net* and VIPER also achieve similar results, showing that Cart Pole is a relatively simple domain compared to the others tested in the main paper. Notably, VIPER achieves a better reward than PW-Net*, but only slightly.

### F.2   ATARI BREAKOUT

Here, we used a pre-trained model from Greydanus (2022). The model uses the Asynchronous Advantage Actor-Critic (A3C) approach. Like Atari Pong, the domain's input state space is raw pixels, and there are six output actions. The model achieves reward for rebounding the ball on screen to hit obstacles above it, and not missing the ball. We limit the number of actions per episode

Table 4: Additional Results: Various baselines are compared to PW-Net. Overall again, only PW-Net consistently performs well.

| Method | Atari Breakout | | Cart Pole | |
|---|---|---|---|---|
| | Reward | Accuracy | Reward | Accuracy |
| PW-Net | **28.14 ± 0.20** | **92.44 ± 0.26** | **200.0 ± 0.00** | **98.69 ± 0.001** |
| VIPER | N/A | N/A | 200.0 ± 0.00 | 54.35 ± 3.03 |
| PW-Net* | 23.41 ± 0.10 | 88.94 ± 0.06 | 197.4 ± 1.40 | 98.69 ± 0.000 |
| k-Means | 00.00 ± 0.00 | 00.00 ± 0.00 | 20.11 ± 9.68 | 14.40 ± 7.77 |
| Black-Box | 26.00 ± 0.81 | N/A | 200.00 ± 0.00 | N/A |

to 1000 to (1) avoid the episode going on too long, and (2) to see which agent performs best as quickly as possible.

The A3C model learned to perform well in the domain, achieving an average reward of around 26.00 across 30 simulations. Following Algorithm 1, we first collected simulation input-output data for 30 simulations, corresponding to 30,000 datapoints for training the PW-Net. To pick the prototypes, this data was partitioned into two six corresponding to each action, and the mean centre of the cluster was used to get the nearest prototype for each action.

We trained five PW-Nets on these data, for 50 epochs, and ran 30 simulations each time to collect the results, and the same was done for PW-Net* and $k$-means. Overall, the PW-Net performed well in the domain with an average reward of 28.14, and an accuracy of 92.44, showing that it approaches the same function as the black-box, but the constraints of using the prototypes means it inevitably learns to reason in a different manner more intuitive for humans.

The other algorithms PW-Net* and k-means achieve 23.41, and 0.00 reward, respectively. This shows that again, PW-Net* performs well, but better performance is achieved using human-defined prototypes and multiple transformations (see Fig. 1). $k$-means performs poorly in this domain, showing the necessity for using transformations in PW-Nets.

Interestingly, the agent here uses a recurrent neural network-based policy. So, this further proves the generality of PW-Nets into different architecture choices.

