# OpenReview forum: "Towards Interpretable Deep Reinforcement Learning with Human-Friendly Prototypes"
_ICLR.cc/2023/Conference — ICLR 2023 notable top 25%_

### Official Review · Reviewer_7Kzz · 2022-10-17

**Confidence:** 4
**Correctness:** 2
**Technical Novelty And Significance:** 3
**Empirical Novelty And Significance:** 2
**Recommendation:** 6

**Clarity, Quality, Novelty And Reproducibility:**

I have the following concern about the paper's clarity.
1. The definition of explanation is not clear. What type of explanation does this paper generate? I assume it is an explanation that explains each individual action.
2. The paper does not specify what exactly the explanation is. It is highlighted features in the input state or weighted prototypes, or something else.
3, The description of the training process is relatively clear. However, the explanation generation process is not specified.

IMHO, the idea of this paper is novel in that it proposes a method to integrate human knowledge into the explanation process. However, since the unclear points mentioned above, it is relatively hard to access the benefits of this new method design. As such, I would rank the overall quality as fair.

Regarding reproducibility, I believe the evaluation in Sections 4 & 5 is reproducible. Still, I have a hard time accessing the reproducibility of the user study due to the unclarity of explanation generation.


**Details Of Ethics Concerns:**

The paper includes an ethical statement to address the potential ethical issues. I agree with the statement and believe it is relatively complete.

**Strength And Weaknesses:**

Strengths:
+ This paper proposes a new explanation method that encodes human-specified prototypes to improve its human explainability.
+ This paper well summarizes the existing literature.

Weaknesses:
- My first major concern about this work is the obtained explanation. After reading the whole paper multiple times, it is still relatively hard to understand what exactly the explanation is. The point I can get is the paper tries to explain the individual actions of an agent. But, It is unclear to me the paper explains the actions through what. For example, most existing works explain individual actions by **highlighting the regions in the input state that are most important to the output action.** In this work, I do not find such a definition. The proposed technique trains the network by predicting the original policy action. After the training, it has prototypical states $S_p$, the weight $W'$, and the encoder $h$. How to generate explanations from these ingredients is not specified. I would highly suggest the authors (1) Give a clear definition of the explanation in this work; (2) Specify how to generate the explanation; (3) Explain the functionality of each component in generating the explanation. (Why we need the prototypes and weight).

- Regarding the technique, I also have the following questions: (1) How to specify the weight $W'$. Should it be dynamically changed in different states and trajectories? (2) What is the influence of wight and prototypical states on the approximation accuracy (MSE) and explanation quality?

- IMHO, the theoretical proof is relatively trivial. The final conclusion is if the similarity is proper, the predicted action is accurate. Since the model is actually learning the proper similarity, this is equivalent to saying if the model $h$ is well trained, the output is accurate. This is obviously true.

- The evaluation is also not that comprehensive. The quantitative evaluation mainly focuses on measuring the difference between the original policy network and the predicted one. This is only the first step. Two networks give the same output may be based on different evidence. The paper misses a quantitative evaluation of the explanation of fidelity or faithfulness. That is, whether the given explanation truly reflects the decision process of the neural policy. The authors could refer to existing evaluation methods for how to design and conduct such an evaluation (e.g., [A Benchmark for Interpretability Methods in Deep Neural Networks]).

- The user study is not that clearly stated. First, how to define in-distribution and OOD? Second, I would suggest providing an example to demonstrate what is presented to the participants and what are the survey questions.

**Summary Of The Paper:**

This paper proposes a new method to explain a pretrained DRL policy. At a high level, it adds a wrap network on top of the original policy network and learns this wrap network by minimizing the difference between its output and the policy network output. Then, it generates explanations using this wrap network. The paper tried to prove the wrapped network could mimic the policy network outputs and empirically show it. Finally, the paper designs a user study to demonstrate the utility of the obtained explanation.

**Summary Of The Review:**

The paper proposes an interesting and new explanation method for DRL policies. It seems to be more human-understandable. However, it is unclear what explanation is generated from the proposed model and how to generate it. This imposes challenges for readers to understand the benefits of the proposed method. Besides, the theoretical analysis is relatively trivial and non-rigorous. Therefore, I would rank this paper below the acceptance bar.

---

> ### Author Response · Authors · 2022-11-14
> **Author Response (Part 1 of 2)**
>
> We thank the reviewer for noting the novelty of our work and for your suggestions which have improved the paper, we now respond to all your questions and concerns. Note that in order to respond adequately, we have split up this initial response across two comments (this is the first).
>
> ***
>
> *(1) Give a clear definition of the explanation; (2) Specify how to generate the explanation; (3) Explain the functionality of each component.*
>
> Thank you for highlighting all of this, we regret the confusion caused.
>
> 1) Our explanation is a classic case-based reasoning one, where explanations are of the form *“I am going to do action x, because this situation is similar to a previous prototypical situation when I also did action x”* [1]. You are correct that the explanation can be distilled down into a **local explanation** which “explains the individual actions of an agent”. These actions are explained through similarity to the prototypes, for example, if it is very similar to the prototype for “turn right” and half similar to the prototype for "accelerate", then it will turn right a lot and accelerate a little. Lastly, since there are only four prototypes in e.g. the CarRacing domain, PW-Nets can also offer a **global explanation** of the car’s concepts which we used in the user study (see Fig. 3).
>
> 2) Since our model is inherently interpretable, the explanation does not need to be generated, *it is naturally produced* when deciding each action to take. As such, the explanation is 100% faithful [1], and not prone to post-hoc error [6]. The explanation is visualised by (i) getting the similarity score of each prototype to the current input state, (ii) multiplying each score by its corresponding weight(s), and then (iii) summing all prototypes for each output action together to get the final action output value (see Fig. 1 & Algorithm 1). In the context of “Accelerating”, there is only one prototype, so if its similarity score is small, the car will accelerate a little, and the explanation for this is *“The state is slightly similar to the prototype for accelerating, so I am going to accelerate a little”*. When combining prototypes for e.g. Steering, the explanation may be *“The state is very similar to the prototype for turning right, and not similar to the prototype for turning left, so I will turn right a lot now”*. So, these explanations are combined if the car e.g. accelerates a little and turns right a lot. We hope this clarifies better how the explanation is visualised for **local explanations**. Otherwise, again all prototypes can just be presented to a user to give a **global explanation** so OOD failures may be predicted (as we demonstrated in the user study).
>
> 3) First, the human-defined prototypes are well motivated by the literature because recent work is discovering that prototype networks benefit from putting humans in the loop [2], coinciding with other work where more prototypes than what seems necessary is needed in order for good performance [3]. Moreover, this seems backed up by theoretical research showing that interpretable representation learning benefits from inductive biases [4]. Second, the weight matrix guarantees that each prototype is associated with intuitive action(s), if *W’* was learned then each prototype could correspond to multiple actions which is bad, e.g. we don’t want the prototype for “Accelerate” to contribute to the output action “Braking”. Third, the multiple transformations are necessary because as we show with *k*-means and PW-Net\*, no transformations, or just one, are not enough to learn the function we need.
>
> As an aside, it would be trivial to add a saliency-map explanation on top of PW-Nets if desired.
>
> **Revision:** We defined explicitly what the explanation is, how to generate them, and the precise functionality of the weight matrix and prototypes in Section 3.2.
>
> ***
>
> *(1) How to specify the weight W′. Should it be dynamically changed? (2) What is the influence of weight and prototypical states on MSE and explanation quality?*
>
> (1) The weights need to be specified by human domain expertise. This seems necessary when considering recent work as discussed above [2,3,4]. So, we decided to leverage human guidance here in a domain which is intuitive for humans, where it makes sense to hand-define them, rather than using indirect learning which may not work. The weight matrix should *never* be changed either in training or testing, it is permanently frozen.
>
> (2) The weight matrix and exact prototypes used have little effect on approximation accuracy (MSE). You can learn the weights, or manually define them, either way you can learn a well performing PW-Net. However, they have a massive effect on *explanation quality*. For example, we force only one prototype to be used for braking, and by manually defining the weight matrix, we don’t allow it to contribute to the action accelerating (which would make no sense to humans).
>
> **Revision:** We discussed this in Section 3.2.

---

> > ### Author Response · Authors · 2022-11-14
> > **Author Response (Part 2 of 2)**
> >
> > *The theoretical proof is relatively trivial.*
> >
> > The “Performance Guarantees” Section was included because no matter how many empirical comparisons we perform, it is never proof that it will generalise fully. Therefore, although the section is not overly complex, we feel it is necessary in order to give an intuition for why PW-Nets, with manual prototypes, a manual weight matrix, and multiple transformations, can learn to perform on-par with a black-box, which isn’t at all obvious since PW-Nets are quite novel in this regard. Furthermore, the analysis reveals why some architectural features are important, such as using 2 prototypes for positive/negative actions, and multiple transformations; this insight might not be obvious without such analysis. If the reviewer agrees, we would rather leave this section to offer a clear intuition and guarantee to those interested in PW-Nets as to why they will work, and to give more confidence in the empirical evaluation.
> >
> > ***
> >
> > *The evaluation is not that comprehensive.*
> >
> > We regret this confusion, but **our goal is not to perfectly mimic the black-box policy**, in fact, our theoretical discussion illustrates how this is actually impossible to do. Our goal is to learn a new policy (based on the original) which uses the human-defined prototypes, thus making the explanations inherently faithful and fidelity issues non-existent. MSE and Accuracy were included because they are part of the optimisation (along with the interpretability constraints), and the agent should be “approaching” the same predictions as the black-box, but never perfectly imitating it. MSE and Accuracy mostly sanity checks.
> >
> > **Revision:** We make it clear throughout the paper that we are not trying to perfectly mimic the original black-box, we are trying to build a new policy which “reasons” with the human-defined concepts, and thus explains itself naturally, **so there are no issues with explanation fidelity**.
> >
> > ***
> >
> > *The user study is not clearly stated.*
> >
> > OOD here refers to situations that involved either a type of road not seen in the original domain (such as a t-junction), or a red obstacle (which is also not in the original domain, but was added in the open-source library that we use). We added the full set of materials in Figure 5, and what was shown to users in Figures 6 and 7 to make this extremely clear. Thank you for making us aware of this.
> >
> > **Revision:** We expanded upon the definition of OOD and ID in the Materials paragraph in Section 6. In addition, we included an example of what the users saw for each question in the appendix (Figures 6 and 7), along with all the ID and OOD materials used to be 100% clear (in Figure 5).
> >
> > ***
> >
> > *...since the unclear points mentioned above, it is relatively hard to access the benefits of this new method design...*
> >
> > We hope these points are now clarified, if not please let us know. As prototype methods are now extremely popular in XAI [5], and PW-Nets appear to be the first such method which doesn’t lose performance, and our user study shows the first practical benefits of such methods, we hope the reviewer agrees we have shown the benefit of this new method.
> >
> > ***
> >
> > *Regarding reproducibility, I believe the evaluation in Sections 4 & 5 is reproducible. Still, I have a hard time accessing the reproducibility of the user study due to the unclarity of explanation generation.*
> >
> > Thank you for noting the reproducibility of Sections 4 and 5. We assure the reviewer there are no issues reproducing the user study; we have given the precise surveys shown to all four groups in the supplementary material. Moreover, we are happy to share the actual online surveys with other researchers if they wish to reproduce the results at any time without having to rewrite the surveys (we note this in Appendix E now). We are confused by how the reviewer lacks confidence that the user study is not reproducible? We are extremely confident the results will reproduce. Please let us know if we misunderstand anything here, thank you for your time.
> >
> > ***
> >
> > [1] Rudin, Cynthia, et al. "Interpretable machine learning: Fundamental principles and 10 grand challenges."
> >
> > [2] Bontempelli, et al. Concept-level Debugging of Part-Prototype Networks.
> >
> > [3] Li, et al. Deep learning for case-based reasoning through prototypes: A neural network that explains its predictions
> >
> > [4] Locatello, et al, May. Challenging common assumptions in the unsupervised learning of disentangled representations
> >
> > [5] Chen et al. This looks like that: deep learning for interpretable image recognition
> >
> > [6] Slack, D.et al, February. Fooling lime and shap: Adversarial attacks on post hoc explanation methods.

---

> > > ### Comment · Reviewer_7Kzz · 2022-11-17
> > > **Thanks for the response and revision.**
> > >
> > > Thanks to the authors for the response and revision. It was able to clarify most of my concerns. I raise the score to 6. Regarding the theoretical part, I think a deeper analysis could be under what situation the model $h$ could be guaranteed to be well-learned. This is challenging and can be taken as future work.

---

### Official Review · Reviewer_EkdB · 2022-10-18

**Confidence:** 4
**Correctness:** 3
**Technical Novelty And Significance:** 3
**Empirical Novelty And Significance:** 3
**Recommendation:** 8

**Clarity, Quality, Novelty And Reproducibility:**

The paper is easy to follow, and clear, however, the narrative could be improved. Also, all the source code required to reproduce the results are provided in the supp material which is nice.


**Strength And Weaknesses:**

Following are suggestions, questions, and concerns that I had while reading the paper.

1- It is mentioned in the paper that “This paper builds upon recent work building prototype-based neural networks for interpretable supervised learning.” However, from [1] and [2] we know that borrowing such methods from other learning approaches could often lead to misunderstanding and faulty insights, since they’re not inherently designed for RL. How does this work address such issues?

2- Related work section needs a better discussion. There’re a lot of works missing that could improve the narrative of the paper, and better describe its goals and motives. A few examples of such papers are: [1], [2], [3], [4]

3- Since the proposed method is model agnostic, it’d be good to have results of other black-box pre-trained models in the paper for each particular environment. For example, in car racing, other than PPO, another policy could be studied. Also, providing results on more environment could help to verify the generalizability and robustness of PW-Net.

4- Doesn’t PW-Net change the behavior of the original black-box pre-trained policy? If yes, how can its interpretations be trusted to reflect the original policy? If no, why there’s a performance difference according to the Table 1?

[1] M. H. Danesh, A. Koul, A. Fern, S. Khorram. Re-understanding Finite-State Representations of Recurrent Policy Networks

[2] A. Koul, A. Fern, S. Greydanus. Learning finite state representations of recurrent policy networks.

[3] A. Mott, D. Zoran, M. Chrzanowski, D. Wierstra, D. J. Rezende. Towards interpretable reinforcement learning using attention augmented agents

[4] A. Atrey, K. Clary, D. Jensen. Exploratory not explanatory: Counterfactual analysis of saliency maps for deep rl

**Summary Of The Paper:**

This paper proposes a new post-hoc explainable method, called PW-Net, to provide insights into the workings of deep RL policies. The main advantage of the proposed method is to use human-friendly prototypes to make the decision-making process of an RL agent clear. Also, a user study is included to support the main claims of the paper.

**Summary Of The Review:**

See above.

---

> ### Author Response · Authors · 2022-11-14
> **Author Response**
>
> We thank the reviewer for noting the advantage of our method, and overall supporting acceptance, we now respond to all your suggestions, questions, and concerns:
>
> ***
>
> *1- It is mentioned in the paper that “This paper builds upon recent work building prototype-based neural networks for interpretable supervised learning.” However, from [1] and [2] we know that borrowing such methods from other learning approaches could often lead to misunderstanding and faulty insights, since they’re not inherently designed for RL. How does this work address such issues?*
>
> Although we borrowed the basic idea of prototype-based explanation from supervised learning, PW-Nets are specifically designed for RL, and are quite different from their “supervised learning counterparts” in several regards. Firstly, we had to use a separate learned representation space for each prototype to perform well (note prior work in supervised learning only used one). Secondly, we defined the prototypes manually, to guarantee intuitive concepts. Lastly,  we manually defined a fixed weight matrix to make the contribution of each prototype intuitive (e.g., we don’t want the prototype for accelerating to contribute to the action braking), note prior work “fine tunes” these weights. Hence, although we built upon this prior work, we had to radically modify it for RL, so the Reviewer’s concerns are unlikely to be an issue.
>
> In addition, from a quantitative perspective, the tests for PW-Net* in the paper (which closely represents the prototype-XAI methods from supervised learning we were inspired by) shows how the prior literature from supervised learning can’t generalise into RL as you are rightly concerned about, and how our method PW-Net overcomes the issues to adapt them into RL by developing an entirely new method which performs well in all tests.
>
> ***
>
> *2- Related work section needs a better discussion. There’re a lot of works missing that could improve the narrative of the paper, and better describe its goals and motives. A few examples of such papers are: [1], [2], [3], [4]*
>
> Thank you for making us aware of these.
>
> **Revision:** We added these citations to improve the narrative of the Related Work and Introduction sections.
>
> ***
>
> *3- Since the proposed method is model agnostic, it’d be good to have results of other black-box pre-trained models in the paper for each particular environment. For example, in car racing, other than PPO, another policy could be studied. Also, providing results on more environment could help to verify the generalizability and robustness of PW-Net.*
>
> We were unsure how many comparisons to include, so we opted for continuous, discrete, and pixel/symbolic state spaces to cover as much ground as possible. Rather than including even more comparisons (which are just empirical), we opted to dedicate space giving an intuition for why the method should always generalise between agents with our “Performance Guarantees” section. However, we agree that more would not hurt.
>
> **Revision:** We included additional PW-Net results in Appendix F for Atari Breakout and Cart Pole, the results converge with the results of the main paper. The code to reproduce these results was also uploaded. Note the Breakout model uses a recurrent neural network policy, showing PW-Nets also generalise to these models, thanks for giving us an opportunity to showcase this.
>
> ***
>
> *4- Doesn’t PW-Net change the behaviour of the original black-box pre-trained policy? If yes, how can its interpretations be trusted to reflect the original policy? If no, why there’s a performance difference according to the Table 1?*
>
> Sorry for the confusion, whilst we did evaluate how accurately PW-Nets mimicked the black-box policy, it was just because that is half of the optimisation problem (the other half being the interpretability constraints), and seeing it “approach” the black-box predictions is just a sanity check the first part of the optimisation is working. Because the PW-Net is jointly learning to (1) mimic the black box, and (2) learn to reason with the human-concepts, it will never perfectly imitate the black-box. Indeed because we are “forcing” it to use the human-defined concepts, it shouldn’t exactly mimic the black-box agent and we don’t want it to, we want it to reason in a wholly new (albeit similar) way, which uses the human-prototypes directly in every decision.
>
> **Revision:** We have made it clear throughout the paper that we are not trying to mimic the black box perfectly, but use it as an additional signal whilst learning to use the human-defined concepts in predictions.
>
> ***
>
> *The paper is easy to follow, and clear, however, the narrative could be improved.*
>
> Thank you for the insight here, we have addressed all your suggestions to make the narrative clearer.

---

> > ### Comment · Reviewer_EkdB · 2022-11-17
> > **Response to authors**
> >
> > I would like to thank the authors for addressing my concerns about the paper. Their additional results improve the significance of the revised paper, thus I would like to increase my score from 6 to 8. Thanks.

---

### Official Review · Reviewer_VRxv · 2022-10-18

**Confidence:** 5
**Correctness:** 3
**Technical Novelty And Significance:** 3
**Empirical Novelty And Significance:** 3
**Recommendation:** 8

**Clarity, Quality, Novelty And Reproducibility:**

The work is clear, and relation to prior work is relatively clearly stated and explained. The user study was not overly clear from a reading of the paper, and required inspection of the source materials to get a better understanding of what participants saw and what their response options were.

While prior work in the field has used exemplars or prototypes for explainability, this work is novel in its use of a hand-defined mapping function from human-specified prototypes to action output, and the method is a valuable contribution to the community.

The work appears to be reproducible and code is included, though I have not tried to run the code to confirm results. The user study materials are included and, with significant effort to rewrite every question, the study could be repeated to confirm those results as well.

**Strength And Weaknesses:**

Strengths:
* The PW-Net is explained clearly and is easy to follow.
* Experimental results verify that black-box policy performance is preserved with the PW-Net wrapper and output mappings.
* Human evaluation shows that the prototypes and PW-Net explanations improve user trust enhance user's abilities to predict policy behavior.

Weaknesses:
* Much of the method must be hand-designed by humans. This is discussed in the paper as a positive, because prior work that simply learns prototypes does not always learn human-usable prototypes, but hand-designing mappings and prototypes is a non-trivial exercise, particularly for something like a 7-DoF robot arm.
* The PW-Net assumes that learned representations are fixed, so any amount of fine-tuning for the policy would require additional fine-tuning for all learned prototypes and embedding functions.
* The prompts and instructions in the user study vary significantly from the control to the proposed method. It is not clear whether human users are rating the control as less trustworthy because the PW-Net prototypes are significantly more useful or explainable, or because participants are being explicitly told that the control has seen "MILLIONS of situations you are NOT being shown." Apart from that very bold text prompt at the top of every question, the two surveys appear to be identical. It therefore seems possible that the user study results are not the result of more interpretable prototypes or a useful mapping function, but just because users are being reminded that one method is less trustworthy than another. Intuitively, the method should improve trust and accuracy, as is reported here, though recent work on explainability methods with humans finds that this is not always the case [1, 2, 3, 4].

[1] Poursabzi-Sangdeh, Forough, et al. "Manipulating and measuring model interpretability." Proceedings of the 2021 CHI conference on human factors in computing systems. 2021.

[2] Silva, Andrew, Mariah Schrum, Erin Hedlund-Botti, Nakul Gopalan, and Matthew Gombolay. "Explainable Artificial Intelligence: Evaluating the Objective and Subjective Impacts of xAI on Human-Agent Interaction." International Journal of Human–Computer Interaction (2022): 1-15.

[3] Peter Hase and Mohit Bansal. 2020. Evaluating Explainable AI: Which Algorithmic Explanations Help Users Predict Model Behavior?. In Proceedings of the 58th Annual Meeting of the Association for Computational Linguistics, pages 5540–5552, Online. Association for Computational Linguistics.

[4] Andrew Anderson, Jonathan Dodge, Amrita Sadarangani, Zoe Juozapaitis, Evan Newman, Jed Irvine, Souti Chattopadhyay, Matthew Olson, Alan Fern, and Margaret Burnett. 2020. Mental Models of Mere Mortals with Explanations of Reinforcement Learning. ACM Trans. Interact. Intell. Syst. 10, 2, Article 15 (June 2020), 37 pages. https://doi.org/10.1145/3366485


**Summary Of The Paper:**

The paper proposes a new method, Prototype-Wrapper-Network (PW-Net) to offer explainability of pre-trained RL policies by training a new set of embedding functions on top of the existing feature representation for a learned policy. These embedding functions are each associated with a given prototype/exemplar for salient actions in a given domain. The fine-tuning process learns to map from the learned policy's representations into similarity functions for each prototype, and the final output action is chosen as a pre-defined weighted combination of similarity scores for each prototype. By having direct access to similarity scores for different actions in the domain, and by pre-defining the mapping from prototypes-to-actions, humans are given the opportunity to better understand actions taken in a domain. The PW-Net is evaluated to show that it does not reduce overall performance for pre-trained policies, and a user-study shows that humans can use the given prototypes to better understand how the policies work (and to trust the method more).

**Summary Of The Review:**

The paper contributes an explainability method based on human labels and action prototypes, offering useful explanations for output actions that are verifiable through known prototypes and similarity scores. While the method is sound and the performance results are convincing (i.e., no performance drops with the PW-Net), the user study design is questionable and it is not clear that the trust and accuracy ratings from participants can be fully attributed to the PW-Net rather than simply to the instructions of the study.

I advocate for acceptance on the basis of the method, but I would like to see a clearer description of the user-study in the main paper and a discussion of possible confounds for the results (such as the instructions telling participants that one method saw millions of other samples). If the study cannot be appropriately defended, I would even advocate removal of the study, on the basis that it is likely not studying explainability, but is instead studying the effects of a text prompt. With clearer explanation of the study or removal of the study, I will increase my score.

---

> ### Author Response · Authors · 2022-11-14
> **Author Response**
>
> We greatly thank the reviewer for supporting the acceptance of our work, noting the novel algorithmic contribution, and considering an increase in their score. We now address all questions and concerns:
>
> ***
>
> *Much of the method must be hand-designed*
>
> Indeed it can be a non-trivial exercise, but we are focusing on domains humans understand well and it makes sense to simply hand-define them. In the Car Racing and Atari Pong domains, it was relatively trivial to define the prototypes and weights as we did, and PW-Nets are not particularly sensitive to the precise prototypes chosen, so performance is rarely an issue.
>
> Whilst it can be laborious, recent work is discovering that prototype networks themselves benefit from putting humans in the learning loop to avoid learning bad, uninterpretable prototypes that worsen performance [1]. This coincides with other work on prototype networks which need more prototypes than seems intuitively necessary (i.e., more than 10 in MNIST) [2]. Moreover, theoretical work is suggesting that learning interpretable representations is perhaps impossible without some biases [3]. Hence, to force the AI to reason successfully with human-defined concepts, the literature is suggesting it is necessary to define the prototypes and weights as we have.
>
> ***
>
> *The PW-Net assumes that learned representations are fixed, so any amount of fine-tuning for the policy would require additional fine-tuning for all learned prototypes and embedding functions.*
>
> Correct, however training the PW-Net is extremely fast, even just using a simple CPU, training on e.g. the Car Racing domain only takes < 1hr. So, practically speaking, this won’t be a huge issue.
>
> ***
>
> *User Study Issues*
>
> Thank you for helping us better present this study. We spent a great deal of time designing the study with an appropriate power analysis, pilot investigations, and attempting to control between groups as closely as possible (which apart from the text prompt in one group we thank the reviewer for noting we seemed to do well), all the while aiming to evaluate a meaningful application.
>
> Regarding the text prompt, we note that the reviewer only directly quoted the “training data” control group as a concern, whilst the other two controls (i.e. the “Non-human friendly”, and the “black-box” groups) seemed not to be a concern? Although there is also extra text in the “black-box” group in Step 2 of the materials [see Fig. 6(B)], this is to control for the fact that users in this group are not seeing 4 images, so to control the cognitive load, additional information is required in the group. We emphasised the “millions of simulations” because the users are not ML experts, and some proxy for achieving a “good reward” and solving the environment needs to be communicated without ML parlance. Since (as far as we know), there are no studies evaluating the effect of recent prototype-based explanation algorithms, we would rather keep the study in an effort to not let human evaluation lag behind algorithmic progress in the field if the reviewer agrees these remaining groups are ok.
>
> To explain our rationale for the text prompt in the “training-data” group, note this group does not have to associate the images with actions (i.e., there is no text heading above the images like the experimental group, see Fig. 7A/B), so the extra text you are concerned with was designed to control for this. However, we understand your concern, and in an earnest response, we decided to remove the “training data” group from the main paper, and wrote a clearer description of the user-study in the main paper discussing how it was controlled between the remaining groups. In addition, the “training-data” group was moved to Appendix E, with a discussion on the possible confounding factor of the text prompt you mention. Note that because of the between-subjects design, the power analysis still holds in this setup, so the study is still valid. We hope this revision addresses the reviewer’s concerns adequately, if not, please let us know and we are happy to do more.
>
> **Revision:** (1) We removed the “training data” group from the main paper, and moved it to the appendix, discussing the possible confounding factors and presenting Figures of the materials shown to each group to make the design clearer (i.e., Figs 5, 6, and 7). (2) A revised explanation of the user study was written in Section 6, paying special attention to how it was controlled across groups.
>
> ***
>
> *The user study materials are included and, with significant effort to rewrite every question, the study could be repeated.*
>
> Please note we are more than happy to share our surveys so people do not have to rewrite them from scratch.
>
> ***
>
> [1] Bontempelli, A.et al. Concept-level Debugging of Part-Prototype Networks.
>
> [2] Li, O.et al Deep learning for case-based reasoning through prototypes
>
> [3] Locatello, F. et al. Challenging common assumptions in the unsupervised learning of disentangled representations

---

### Official Review · Reviewer_KNoV · 2022-10-25

**Confidence:** 4
**Correctness:** 3
**Technical Novelty And Significance:** 3
**Empirical Novelty And Significance:** 2
**Recommendation:** 8

**Clarity, Quality, Novelty And Reproducibility:**

The paper is well-written and has a good structure. Regarding novelty, it is the first deep reinforcement learning method based on prototypes. The authors of the paper provide both the code and the user surveys to be straightforward to reproduce the results.

**Strength And Weaknesses:**

Strengths:
- First prototype-based deep reinforcement learning technique
- Can be added to any pre-trained agent
- Experiments fit the task by comparing the proposed method to the black-box agent
- Evaluated the technique in a user study

Weaknesses:
- Experiments section lacks some explanations of the evaluation metrics and results (see questions Q2-Q4)
- Needs hand-crafted prototypes: my concern is the ease-of-use of the method, as it needs extra work from humans to define the prototypes (considering all actions with multiple prototypes) and also the weight matrix. End-to-end learning is usually preferred because it doesn’t need extra hand-crafted elements.

Questions and suggestions for the authors:
Q1: Figure 1.: indices of h, z and p. Even though it is highlighted in the text that the model uses separate mapping, the image of the network shows the same indices (i,j).

Q2: What is the reason for the different evaluation metrics (MSE vs Accuracy) of discrete and continuous action spaces?

Q3: There are no results of the VIPER in Car Racing and Atari Pong. Why?

Q4: As your technique doesn’t give any advantage to the trained agent, how could the PW-Net achieve slightly better results in the experiments section?


**Summary Of The Paper:**

The paper focuses on how to construct a wrapper model that is applicable to any arbitrary trained agents which leads to interpretability improvement in deep reinforcement learning. Their technique is called Prototype Wrapper Network (PW-Net) with the key concept of using human knowledge for prototypes and mapping between prototypes and actions.

The main contributions and findings of the paper:
- Using prototype-based learning in deep reinforcement learning
- Pre-hoc interpretability, forcing the agent to learn the human-designed prototypes makes the model inherently explainable
- Performance of the agent doesn’t degrade by using the proposed wrapper
- User trial to measure user’s predictions (failures and successes) of different models

**Summary Of The Review:**

Overall, I vote for accepting. I like this paper because the proposed technique can provide inherent interpretability for deep reinforcement learning agents. My main concern is the extra work and time of creating human-defined prototypes and weight matrix that is needed for the wrapper network.

---

> ### Author Response · Authors · 2022-11-14
> **Author Response**
>
> We thank the reviewer on account of voting for the acceptance of our work. Here, we address specific questions and concerns.
>
> ***
>
> *My concern is the ease-of-use of the method, as it needs extra work from humans to define the prototypes*
>
> Whilst we agree end-to-end learning is usually preferred (indeed we originally experimented with this), we wanted to 100% guarantee that the prototypes would be interpretable, and their contribution to actions make sense (i.e., with the weight matrix), which in reality, requires some degree of human oversight, especially in truly sensitive domains [1]. We chose to focus on driving, as it is a well understood domain by humans where defining concepts (e.g., turn left) and a weight matrix is relatively trivial and makes sense.
>
> Recent work suggests that it is necessary to incorporate humans into the mix in order to maximise the effectiveness of prototype networks [3]. Indeed, work on prototype networks has shown that more prototypes than a human might deem “intuitively necessary” is needed for good performance, even on simple domains like MNIST [4]. This seems backed up by work that shows learning human-interpretable disentangled representations is both empirically and theoretically impossible without useful inductive biases or human supervision [2]. Hence, to guarantee the AI will align with human-intuition, the literature is suggesting it is necessary to define the prototypes and weights as we have, and notably this doesn’t hinder performance at all.
>
> ***
>
> *Q1: Figure 1*
>
> We used the indices (i,j) to imply that they could have any value, but in the figure they would be h_1,1, h_1,2 etc…
>
> **Revision:** We clarify this in the figure caption.
>
> ***
>
> *Q2: What is the reason for the different evaluation metrics (MSE vs Accuracy) of discrete and continuous action spaces?*
>
> The training is optimising two things: (1) PW-Net is learning to closely mimic the black-box policy actions, and (2) it is learning to reason in a new way using the human-defined prototypes. We used MSE and Accuracy to gauge if the predictions of PW-Net (i.e., the “predictions” in supervised learning), are getting closer to the predictions of the agent (i.e., the “labels” in supervised learning). As MSE is usually used to evaluate regression domains, and Accuracy to evaluate classification domains, they seemed the best choice when evaluating the continuous and discrete actions spaces, respectively. However, we are quite open to other metrics, does the reviewer have anything specific in mind? These metrics are mostly a sanity check to see if part (1) of the optimisation is working.
>
> As an aside, please also recall that we are not trying to perfectly match the predictions of the agent with PW-Net, but rather use it as a “guide” whilst learning a new policy with PW-Net.
>
> **Revision:** We discussed the rationale for the metrics in Sections 4 and 5.
>
> ***
>
> *Q3: There are no results of the VIPER in Car Racing and Atari Pong. Why?*
>
> Good question, VIPER does not work with state spaces which are raw pixels, so it unfortunately couldn’t be applied to the deep RL domains of Car Racing and Atari Pong. VIPER only works in symbolic domains. Although the original VIPER paper did do a test on Atari Pong, it manually extracted a 7-dimensional symbolic state space from the images, rather than using the raw pixels. We note this in the revised manuscript, thanks for pointing that out.
>
> **Revision:** We note in Section 4.2 why VIPER is not tested on the domains where state spaces are raw pixels.
>
> ***
>
> *Q4: As your technique doesn’t give any advantage to the trained agent, how could the PW-Net achieve slightly better results in the experiments section?*
>
> Our intuition is that PW-Net acts as a form of regularisation (due to relying on e.g. only four prototypes, when most prototype literature use 100’s), and this can help them generalise better in some circumstances. Note these results changed slightly after correcting a version issue in our code (most notably on Atari Pong due to its ~5% stochastic policy), but PW-Nets still do better than the black-box in several domains tested.
>
> ***
>
> [1] Rudin, C., et al., 2022. Interpretable machine learning: Fundamental principles and 10 grand challenges. Statistics Surveys, 16, pp.1-85.
>
> [2] Locatello, F., Bauer, S., Lucic, M., Raetsch, G., Gelly, S., Schölkopf, B. and Bachem, O., 2019, May. Challenging common assumptions in the unsupervised learning of disentangled representations. In international conference on machine learning (pp. 4114-4124). PMLR.
>
> [3] Bontempelli, A., Teso, S., Giunchiglia, F. and Passerini, A., 2022. Concept-level Debugging of Part-Prototype Networks. arXiv preprint arXiv:2205.15769.
>
> [4] Li, O., Liu, H., Chen, C. and Rudin, C., 2018, April. Deep learning for case-based reasoning through prototypes: A neural network that explains its predictions. In Proceedings of the AAAI Conference on Artificial Intelligence (Vol. 32, No. 1).

---

> > ### Comment · Reviewer_KNoV · 2022-11-21
> > **Reviewer Response**
> >
> > Thank you to the authors for answering our questions. With the clarifications and revision we will raise the score to 7.

---

### Author Response · Authors · 2022-11-14
**Dear Reviewers and AC**

We would like to thank the reviewers for taking the time to read our paper and give constructive feedback. We have taken seriously the suggestions of all reviewers and either amended the manuscript accordingly to address the concern, or discussed it here in our response.

Firstly however, we thank the reviewers for their positive comments, and overall vouching for acceptance, Reviewer-KNoV stated *“Overall, I vote for accepting. I like this paper because the proposed technique can provide inherent interpretability for deep reinforcement learning agents.”*, Reviewer-VRxv said *“I advocate for acceptance on the basis of the method… this work is novel… and the method is a valuable contribution to the community”*, Reviewer-EkdB pointed out that *“The main advantage of the proposed method is to use human-friendly prototypes to make the decision-making process of an RL agent clear. Also, a user study is included to support the main claims of the paper”*, lastly Reviewer-7Kzz said *“The paper proposes an interesting and new explanation method for DRL policies”*.

We are truly grateful for these (and other) encouraging comments made.

Beyond the positive reviews, we are grateful for many of the suggestions provided and have incorporated them into the latest revision to improve the manuscript. Our changes include:

* Due to concerns from Reviewer VRxv about the “training data” group from the user study being less controlled than the other three groups, it was removed from the main paper and moved to the Appendix E with a discussion on these concerns to note the possible confounding factor. Due to the between-subjects design, the power analysis still holds for the remaining three groups, so the design is still valid.
* In response to a common thread from Reviewers KNoV, VRxv, and 7Kzz, we expanded our discussion on why it is necessary (and good) to human-define the prototypes and weight matrix W’ in Section 3.2.
* In response to Reviewers VRxv and 7Kzz, we expanded the user study in Section 6 to give more details, and the materials with example survey questions were added in Appendix E (Figures 5/6/7), overall to facilitate a better description of the study.
* In response to Reviewer 7Kzz, the method is better discussed with a clearer description of what the explanation is, the functionality of each component, and how to visualise it in Section 3.2.
* Regarding Reviewer EkdB’s suggestions related to the paper’s narrative, all recommended citations were included to give the paper a better narrative in Sections 1 and 2.
* Regarding a concern from Reviewers 7Kzz and EkdB, we clarified across the paper how we are not trying to perfectly match the predictions of the agent with PW-Net, but rather use the agent as a “guide” whilst learning a new policy with PW-Net that uses human-interpretable prototypes for its reasoning. So, issues of fidelity between PW-Net and the black-box raised by these reviewers are not an issue. Apologies for this confusion.
* Upon the suggestion of Reviewer EkdB, we included some additional results in Appendix F to further prove the generality of PW-Nets. Specifically we included results for Atari Breakout (a deep RL domain), and Cart Pole (symbolic domain). The results converged with all others and we uploaded the code.
* Lastly, as an aside, we briefly note that we corrected a version issue in our code (and re-uploaded the correct version). This slightly changed the results, but for clarity, we highlight the peripheral edits that were made. First, in the Lunar Lander domain, our new findings corroborate trends from other results, wherein PW-Net outperforms PW-Net*. Second, while PW-Net remains the best method on the Atari Pong domain, and performs on par with the black-box when using a deterministic policy, the results reported here are slightly lower due to using a stochastic policy which is common practice [1,2].


[1] Ilya Kostrikov. Pytorch implementations of asynchronous advantage actor critic. https://github.com/ikostrikov/pytorch-a3c, 2018

[2] Beḩet Ş ent ̈urk. https://github.com/bhctsntrk/openaipong-dqn, 2022. URL https://github.com/nikhilbarhate99/Actor-Critic-PyTorch

---

### Decision · Program_Chairs · 2023-01-20

**Decision:**

Accept: notable-top-25%

**Justification For Why Not Higher Score:**

Prototype-based explanation has been studied for supervised learning counterparts.

**Justification For Why Not Lower Score:**

Nonetheless, the paper made several well-justified adaptation to enable first use case of prototype-based explanation for deep reinforcement learning.

**Metareview: Summary, Strengths And Weaknesses:**

The paper proposes Prototype-Wrapper-Network (PW-Net) to post-hoc explain pre-trained RL policies by learning to map the learned policy's representation into similarity functions to human designed prototypes, and the output action by the learned policy is explained as a pre-defined weighted combination of similarities to the prototypes. Reviewers are unanimously in favor of accepting the work for its novelty, well execution and the inherent interpretability. Reviewers also raised concerns on domain expertise is required to manually define the prototypes and the weight matrix, but were reasonably convinced through the rebuttal.



**Note From Pc:**

if the above contains the word "oral" or "spotlight" please see: "oral" presentation means -> notable-top-5% and "spotlight" means -> notable-top-25%. As stated in our emails, we are disassociating presentation type from AC recommendations